

# Uncovering the mechanism of anthocyanin accumulation in a purple-leaved variety of foxtail millet (*Setaria italica*) by transcriptome analysis

Yaofei Zhao[*], Yaqiong Li[*], Xiaoxi Zhen, Jinli Zhang, Qianxiang Zhang, Zhaowen Liu, Shupei Hou, Yuanhuai Han and Bin Zhang

Shanxi Key Laboratory of Minor Crop Germplasm Innovation and Molecular Breeding, College of Agriculture, Shanxi Agricultural University, Taigu, Shanxi, China
[*] These authors contributed equally to this work.

Corresponding authors
Yuanhuai Han, hanyuan-huai@sxau.edu.cn
Bin Zhang, Abingood@126.com

## ABSTRACT

Anthocyanin is a natural pigment that has a functional role in plants to attract pollinating insects and is important in stress response. Foxtail millet (*Setaria italica*) is known as a nutritional crop with high resistance to drought and barren. However, the molecular mechanism regulating anthocyanin accumulation and the relationship between anthocyanin and the stress resistance of foxtail millet remains obscure. In this study, we screened hundreds of germplasm resources and obtained several varieties with purple plants in foxtail millet. By studying the purple-leaved B100 variety and the control variety, Yugu1 with green leaves, we found that B100 could accumulate a large amount of anthocyanin in the leaf epiderma, and B100 had stronger stress tolerance. Further transcriptome analysis revealed the differences in gene expression patterns between the two varieties. We identified nine genes encoding enzymes related to anthocyanin biosynthesis using quantitative PCR validation that showed significantly higher expression levels in B100 than Yugu1. The results of this study lay the foundation for the analysis of the molecular mechanism of anthocyanin accumulation in foxtail millet, and provided genetic resources for the molecular breeding of crops with high anthocyanin content.

## INTRODUCTION

Flowers, fruits and seeds generally display different colors in nature to attract animal pollinators or seed dispersers (*Grotewold, 2006*). The coloration is determined by the deposition of pigments, including betalains, carotenoids and anthocyanins. Anthocyanins, which have been studied most, are responsible for the red, blue and purple colors of plant tissues. Additionally, anthocyanins have been found to perform pivotal roles in a plant's response to environmental stresses. Anthocyanin accumulation can be induced by biotic or abiotic stresses including pathogen infection, drought, salt, cold, mechanical

damage and high light intensity (*Albert et al., 2009*; *An et al., 2020*; *An et al., 2019*; *Kim et al., 2017*; *Li et al., 2017*; *Naing et al., 2018*; *Nakabayashi et al., 2014*; *Qin et al., 2020*; *Shen et al., 2017*; *Wang et al., 2018*; *Yong, Zhang & Lyu, 2019*). Anthocyanins have been found to have health-promoting factors which defend against disease, improve immunity, and prevent cardiovascular diseases. Therefore, anthocyanins are of great interest for research and practical applications (*He & Giusti, 2010*; *Lila et al., 2016*).

Anthocyanins are derived from a branch of the flavonoid metabolism pathway in plants. The biosynthesis of anthocyanins is catalyzed by a series of enzymes, such as chalcone synthase (CHS), chalcone isomerase (CHI), flavanone 3-hydroxylase (F3H), dihydroflavonol 4-reductase (DFR), anthocyanidin synthase (ANS), and UDP-3-O-glucosyltransferases (UFGT/3GT). Anthocyanins are water-soluble, synthesized in the cytoplasm, and then transported to the vacuole for storage (*Koes, Verweij & Quattrocchio, 2005*). Anthocyanins are found in the flowers, fruits, seeds, leaves, and stems, and their distribution varies based on plant species, plant tissues, developmental stages and environmental factors. In *Arabidopsis thaliana*, anthocyanins are distributed in the subcutaneous tissue cells of the adaxial side and epidermal cells of the abaxial side of rosette leaves (*Kubo et al., 1999*). In red-leaved ornamental kale (*Brassica oleracea var. acephala* DC) leaves, anthocyanins are distributed in epidermal cells and mesophyll cells adjacent to the epidermis; however, there was no pigment deposition in the internal mesophyll cells. Anthocyanin accumulation varied in young leaves and mature leaves of *Mikania micrantha* under low-temperatures. Young leaves showed high pigment deposition in the adaxial and abaxial sides, whereas mature leaves only had high pigment deposition in the abaxial side. Additionally, the total antioxidant capacity of young leaves was significantly higher than that of mature leaves, while the photosynthetic rate of mature leaves was significantly higher than that of young leaves. Thus, plants can modulate anthocyanin distribution to adapt to different environments (*Zhang et al., 2019a*).

Foxtail millet (*Setaria italica*) is an important $C_4$ cereal and forage crop, with high photosynthetic efficiency and good adaptability to the environment. These advantages, together with the properties of self-pollination, a small genome and extensive genetic diversity make foxtail millet a model plant for the $C_4$ cereal crop with tremendous potential. To further promote the development of functional genomics of foxtail millet, researchers identified a mutant (*xiaomi*) with a similar life cycle and size to Arabidopsis (*Yang et al., 2020*). High quality reference genomes were assembled, a multi-omics database was constructed, and a highly efficient genetic transformation system was established to explore the genetic resources and molecular breeding of cereal grains. Although the biological function and metabolic mechanism of anthocyanins have been extensively studied in some plants such as Arabidopsis, maize (*Zea mays*), tomato (*Lycopersicon esculentum*), and snapdragon (*Antirrhinum majus*), the research in foxtail millet is still in its infancy (*Albert et al., 2021*; *Petroni & Tonelli, 2011*; *Sun et al., 2020*). In this study, purple- and green-leaved foxtail millet varieties were used as experimental materials to analyze the distribution of anthocyanin, physiological indexes, and transcriptome sequencing analysis. The results of our study will help to explore the mechanism of anthocyanin accumulation

in foxtail millet and lay a foundation for the further study of the molecular mechanism of anthocyanin synthesis regulation in foxtail millet.

# MATERIALS & METHODS

## Plant material and growth conditions

The seeds of foxtail millet varieties B100 (purple-leaved variety) and Yugu 1 (green-leaved variety) were sown in an experimental plot located at 37°N, 112°E. The leaves were collected for analysis when plants reached a mature four-leaf stage.

## Morphological observations

The leaves were sliced using two, side-by-side blades for morphoanatomical evaluation. The slices were embedded in water and were then observed under a microscope.

## Measurement of plant pigments

Anthocyanin was extracted from leaves with 0.1% HCl-methanol at 4 °C for 12 h in dark. The absorption values of the extraction were measured at 530 and 700 nm wavelengths using the mixture of supernatant and reagent 1 (50 mmol $L^{-1}$ KCl, 150 mmol $L^{-1}$ HCl, pH = 1.0), followed by the mixture of supernatant and reagent 2 (100 mmol $L^{-1}$ NaAC, 240 mmol $L^{-1}$ HCl, pH = 4.5). The method of calculation was as follows (*Luo et al., 2017*):

Anthocyanin (mg/g FW) = $[(A_{520} - A_{700})_{\text{reagent 1}} - (A_{520} - A_{700})_{\text{reagent 2}}] \times M \times V \times n/(\varepsilon \times m)$.

M: relative molecular weight of anthocyanins, 449.2 g/mol; V: extraction volume; $\varepsilon$: molar extinction coefficient of anthocyanins, $2.69 \times 104$ L/mol/cm; n: dilution times; m: sample weight.

Chlorophyll and carotenoids were extracted from leaves using 80% acetone for 24 h. Then the absorption values were measured from the extraction at 663, 646 and 470 nm wavelengths (*Porra, 2002*). The method of calculation was as follows (V: extraction volume, W: sample weight):

Chlorophyll a (mg/g Fw) = $(12.21A_{663} - 2.81A_{646}) \times V/W \times 1{,}000$

Chlorophyll b (mg/g Fw) = $(20.13A_{646} - 5.03A_{663}) \times V/W \times 1{,}000$

Chlorophyll (mg/g Fw) = chlorophyll a (mg/g Fw) + chlorophyll b (mg/g Fw)

Carotenoid (mg/g Fw) = $[1{,}000A_{470} - 3.27 \times (12.21A_{663} - 2.81A_{646}) - 104 \times (20.13A_{646} - 5.03A_{663})]/229 \times V/W \times 1{,}000$.

The raw data of the pigment measurement is shown in Table S5.

## Total antioxidant capacity assay

Antioxidants were extracted from leaves with 80% methanol. The antioxidant capacity was measured using the ferric-reducing antioxidant power (FRAP) method (*Benzie & Strain, 1996*).

Trolox solutions were prepared using methanol as the solvent of gradient concentrations. The Trolox solutions and FRAP solution were mixed for reaction for 1 h in the dark. Then we measured the absorption value of the mixture at 593 nm wavelength. The standard curve was drawn by taking the Trolox concentration as the $x$-axis and the absorbance as the $y$-axis.

The leaf extracts and FRAP solution were mixed for reaction for 1 h in the dark. Then we measured the absorption value of the mixture at 593 nm wavelength. The antioxidant capacity was assayed according to the standard curve and absorbance. The raw data of this assay is shown in Table S6.

## Measurement of soluble sugar content

The leaves were homogenized with 80% ethanol and were extracted at 80 °C for 30 min. The supernatant was transferred to a fresh, clean tube after centrifugation at 3,500 $g$ for 10 min. The sediment was then extracted twice using the above procedure with 80% ethanol. The supernatant was added into an anthrone-sulfuric acid reagent and was boiled for 10 min. After cooling to room temperature, the absorption value of the extraction was measured at 620 nm (*Leng et al., 2016*). The raw data of this assay is shown in Table S6.

## Transcriptome sequencing and analyses

The total RNA was extracted from leaves using RNAiso Plus (9109; Takara). The RNA preparations were evaluated with agarose gel electrophoresis and were quantified using a microultraviolet–visible fluorescence spectrophotometer.

Three biological replicates of both Yugu1 and B100 were used in RNA-sequencing. The statistical power of this study was calculated using RNASeqPower (https://bioconductor.org/packages/release/bioc/html/RNASeqPower.html) as $P < 0.05$. The quality control of raw data was performed as in a previous study (*Wang et al., 2017*). We retrieved 144,677,546 clean reads, including 43.4 G total bases. The sequencing depth was 100x. The clean data were aligned to the foxtail millet genome (Yugu1, version 2.2: https://data.jgi.doe.gov/refine-download/phytozome?organism=Sitalica{&}expanded=312) using HISAT2 software (*Kim, Langmead & Salzberg, 2015*), and the mapped reads were analyzed by String Tie software (*Pertea et al., 2015*). The abundance of transcripts was normalized using the fragments per kilobase of transcript per million mapped reads (FPKM) method (*Florea, Song & Salzberg, 2013*). Prior to differential gene expression analysis, for each sequenced library, the read counts were adjusted using the edgeR program package with one scaling normalized factor. Differential expression analysis of the two samples was performed using the EBSeq R package. The resulting false discovery rate (FDR) was adjusted using the posterior probability of being DE (PPDE). The FDR $< 0.05$ & |log2 (fold change)| $\geq 1$ was set as the threshold for significantly differential expression. Differential expression analysis of the two conditions/groups was performed using the DESeq R package (1.10.1). DESeq provided statistical routines for determining the differential expression in digital gene expression data using a model based on the negative binomial distribution. The resulting $P$ values were adjusted using the Benjamini and Hochberg's approach for controlling the FDR. Genes with an adjusted $P$-value $< 0.05$ by DESeq were assigned as differentially expressed.

We performed Gene Ontology (GO) enrichment analysis by the GOseq R package method (*Young et al., 2010*). The KEGG database (http://www.genome.jp/kegg/) was used to discover the regulatory pathways of genes (*Kanehisa et al., 2008*). We used KOBAS (*Mao et al., 2005*) to determine the enrichment of DEGs in KEGG pathways.

The raw data of the RNA sequencing was submitted to an SRA database (BioProject accession number: PRJNA777600).

## Quantitative real-time PCR analysis

The total RNA was isolated as mentioned above. The first-strand cDNA was synthesized using the PrimeScript RT reagent kit with gDNA Eraser (RR047A; Takara). qRT-PCR was performed using the Bio-Rad CFX96TM Touch real-time PCR detection system with SYBR Premix Ex TaqTM II (Tli RNaseH Plus) (RR820A; Takara). Transcript levels were normalized to the *SiActin* (Seita.5G464000). The raw data of qRT-PCR is shown in Table S7. The $2^{-\Delta\Delta Ct}$ method was used to calculate the expression level of the genes and the one-way ANOVA was used to identify significant differences (http://vassarstats.net/anova1u.html).

# RESULTS

## Purple-leaved variety B100 accumulated more anthocyanin than YG1 in epidermal cells

In a previous study, we obtained a few foxtail millet varieties with different leaf colors, stems and panicles. We selected two representative varieties B100 (purple-leaved variety) and Yugu1 (YG1, green-leaved variety) to analyze the variation of anthocyanin distribution in foxtail millet. The morphological characteristics showed that B100 presented the visible purple color of the leaves at the seedling stage but the leaves of YG1 were green (Fig. 1A). Furthermore, the leaves of B100 and YG1 showed a different color at the maturation stages (Fig. 1B). The purple color of the purple leaves was deepest in the vein, which implied that anthocyanin was transported through the vascular bundle. Anatomical observations of the leaf revealed that in the leaves of B100, purple pigmentation was present in the adaxial and abaxial epidermal cells as well as adjacent mesophyll cells, whereas the purple pigmentation was not significantly present in the leaves of YG1 (Figs. 1C–1D).

## The content of anthocyanin, not other pigments, was higher in B100 than YG1

We quantitatively analyzed the content of some pigments in B100 and YG1 to examine whether the visible purple was due to an increase in anthocyanin accumulation. The results showed that the anthocyanin content in B100 was significantly higher than YG1 at the seedling stage and the maturation stage (Fig. 2A), while the chlorophyll and carotenoid contents were higher in YG1 than in B100 (Figs. 2B–2C). Furthermore, all three pigments contents were higher at the maturation stage than at the seedling stage and the anthocyanin content was similar at the two stages in YG1.

## B100 displayed enhanced stress resistance

Since anthocyanin synthesis and accumulation are closely related to the stress responses in plants, we explored the functions of anthocyanin in regulating stress resistance in foxtail millet. The total antioxidant capacity analysis was performed using the ferric-reducing antioxidant power (FRAP) method and we established the standard curve (Fig. S1A). B100 was shown to accumulate more anthocyanin and was better able to scavenge free radical and reductive ferrous ions than YG1 (Fig. 3A). Meanwhile, B100 could accumulate more

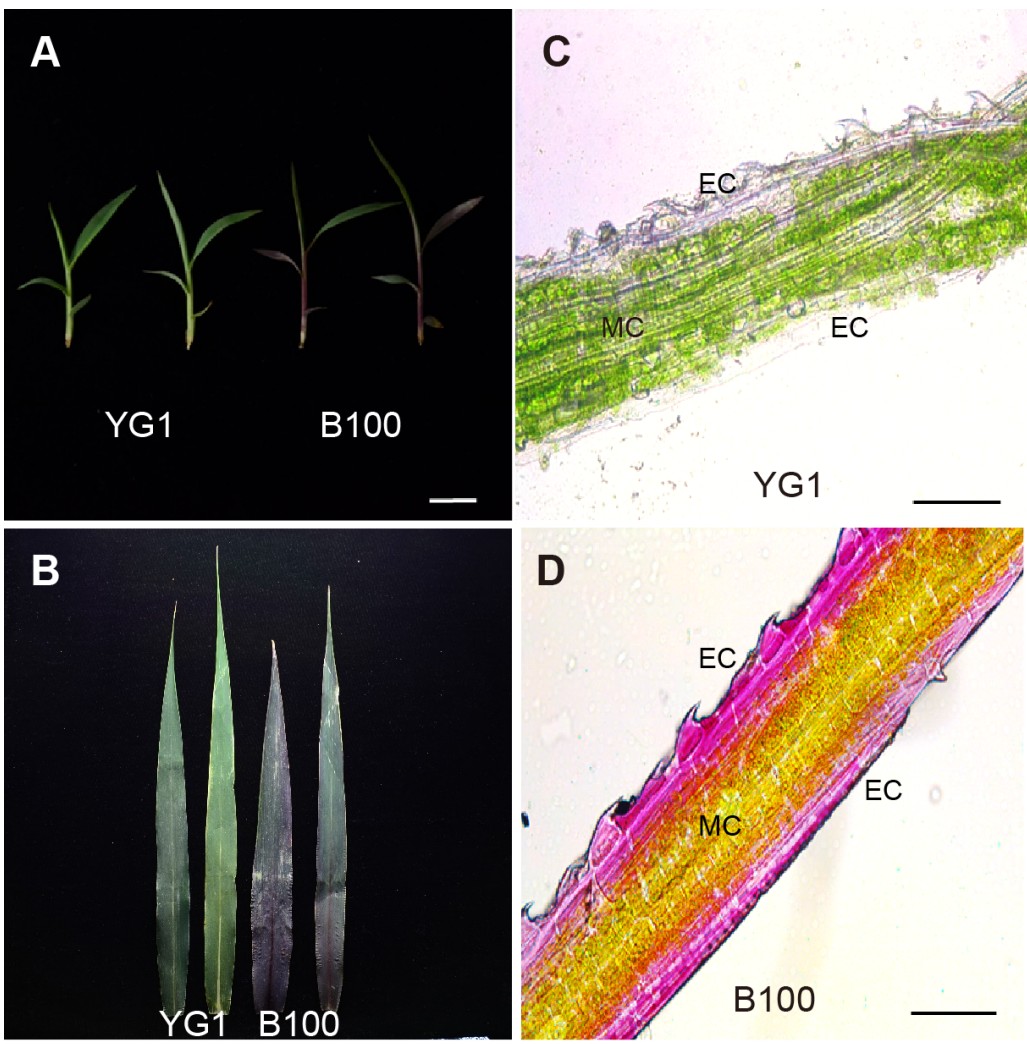

**Figure 1** **Leaf phenotype analysis of YG1 and B100.** (A) Seedlings of YG1 and B100. Bar = three cm. (B) Flag leaves of YG1 and B100 in the maturing stage. Bar = 10 cm. (C–D) Transverse sections of YG1 leaves and B100 leaves, bars = 100 μm. MC, mesophyll cells; EC, epidermal cells.

soluble sugar in both the seedling and maturation stages compared to YG1 (Fig. 3B). The standard curve of the quantitative determination of soluble sugar is shown in Fig. S1B.

## Differentially expressed genes were identified through the analysis of transcriptome characteristics in purple- and green-leaved foxtail millet varieties

We conducted RNA sequencing to elucidate the molecular mechanism underlying the phenotypic and functional differences between the purple- and green-leaved varieties. The total RNA of B100 and YG1 was extracted from the top second leaf at the maturation stage. The quality of the RNA was detected by agarose gel electrophoresis (Fig. S2A), and all RNA of corresponding samples that fulfilled the requirements were used for RNA-sequencing. A total of 144,677,546 clean reads were obtained, with a total base number of 43.4 G and GC

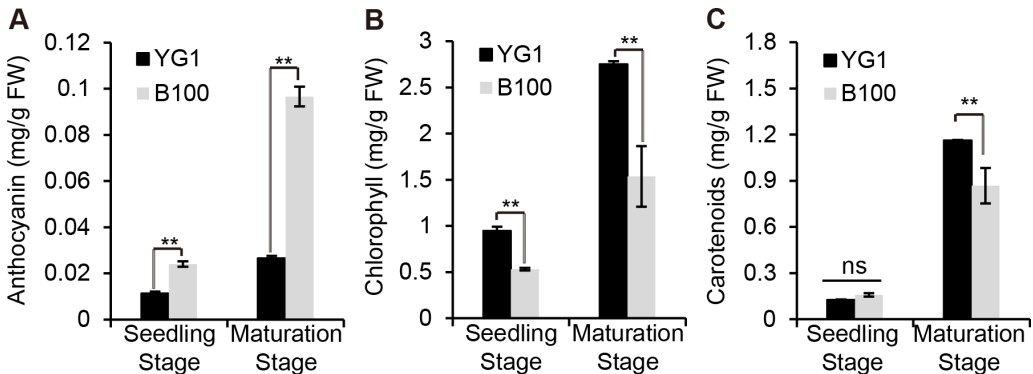

**Figure 2** **Quantitative analysis of plant pigments.** (A–C) The total content of anthocyanin, chlorophyll and carotenoids in leaves of YG1 and B100. Contents were detected at both the seedling and maturing stages. Bars represent means ± SD of three biological replicates. Two asterisks (**) represent $P < 0.01$, ns represents no significant difference ( $n = 3$, ANOVA). FW, fresh weight.

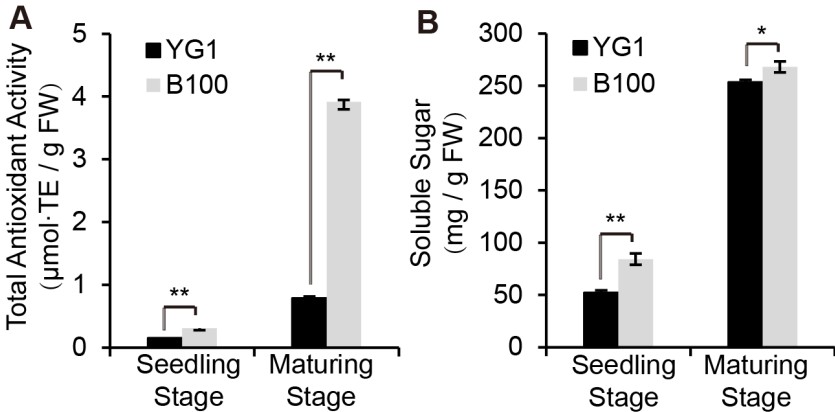

**Figure 3** **Quantitative analysis of physiological indexes to stress resistance.** The total antioxidant activity was determined using the FRAP method (A) and the total content of soluble sugar is shown. The bars represent the means ± SD of three biological replicates. Two asterisks (**) represent $P < 0.01$ ( $n = 3$, ANOVA). FW, fresh weight.

contents were between 58.24% and 59.01%. The quality values of the sequencing data were statistically assessed for each sample of three biological replicates. The Q30 values were all above 92.84%, indicating that the transcriptome data were of high quality and could be further analyzed (Table S1).

The clean reads were sequentially compared with the reference genome (*Setaria italica* v2.2, Yugu1). Approximately 90.38%-91.50% of the clean reads in YG1 were sequentially compared to the reference genome. In the purple-leaved variety B100, 88.05%-89.97% of reads were compared to the reference genome. Among them, the proportion of clean reads with multiple locations was between 1.42% and 1.72%, and the number of reads compared to the positive and negative strands of the genome was similar. Therefore, the selected reference millet genome assembly may meet the needs for analysis (Table S2). We

then obtained 1,184 significant differentially expressed genes (DEGs) of YG1 and B100, in which 598 genes were up-regulated and 588 genes were down-regulated in B100 compared with YG1 (Fig. S2B). A cluster analysis of these genes was conducted by hierarchical clustering. According to the different expression patterns, these DEGs were clustered into three groups. The genes in cluster I were down-regulated in B100, and the genes in cluster III were up-regulated in B100 (Fig. 4).

Subsequently, enrichment analysis of DEGs was performed based on Gene Ontology (GO) analysis and Kyoto Encyclopedia of Genes and Genomes (KEGG) pathway analysis. GO analysis showed that DEGs were significantly enriched in the metabolic process, cell and cell part, binding, and catalytic activity (Fig. 5, Table S8). KEGG pathway analysis revealed that the DEGs were mainly enriched in the metabolic pathways. Among these DEGs, we identified some genes encoding synthetase of flavonoid biosynthesis and phenylpropane biosynthesis which may be related to anthocyanin metabolism (Fig. 6).

### Nine structural genes related to anthocyanin biosynthesis may function in the accumulation of anthocyanin

We screened nine structural genes involved in anthocyanin biosynthesis from the results of the significant differentially expressed genes analysis that were up-regulated in B100 compared to YG1. We performed quantitative real-time PCR to verify the results of RNA sequencing in foxtail millet to determine the major role of these genes. As shown in Fig. 7, the relative expression levels of all nine structural genes in B100 were significantly higher than YG1 at the maturation stage. Among these up-regulated genes, *DFR* (Seita.5G237900), *LDOX-2* (Seita.1G000700) and *AT* (Seita.9G002300) had higher expression levels in B100 than YG1 at both the seedling and maturation stages. With the development of foxtail millet, most of these structural genes displayed an increasing expression level. However, the plant accumulated more transcripts of *PAL* (Seita.1G240200) and *UFGT* (Seita.3G190000) at the seedling stage. All of the nine DEGs in the B100 purple-leaved variety were up-regulated, which was consistent with the results of RNA-sequencing. Moreover, the correlation between qRT-PCR and RNA-sequencing was 0.9625 (Fig. S3). These results showed that the sequencing results had high accuracy and reliability, and may be used for further investigations.

## DISCUSSION

Compared with other crops, foxtail millet has many elite traits, including high tolerance to abiotic stresses and low requirement of fertilizer in cultivation, which make foxtail millet an ideal staple food crop widely cultivated in semi-arid regions or infertile land (*Muthamilarasan & Prasad, 2021*). Foxtail millet is also an important crop to ensure food and nutritional security in the face of a rapidly expanding world population (*Peng & Zhang, 2020*). The purple-leaved B100 displayed an enhanced resistance to stresses, which may be due to anthocyanin accumulation. Anthocyanins can improve the antioxidant capacity of plants. Under various stresses, plants produce reactive oxygen species (ROS) which damage cells. In order to survive under stress, ROS and damage to cells, excessive ROS must be

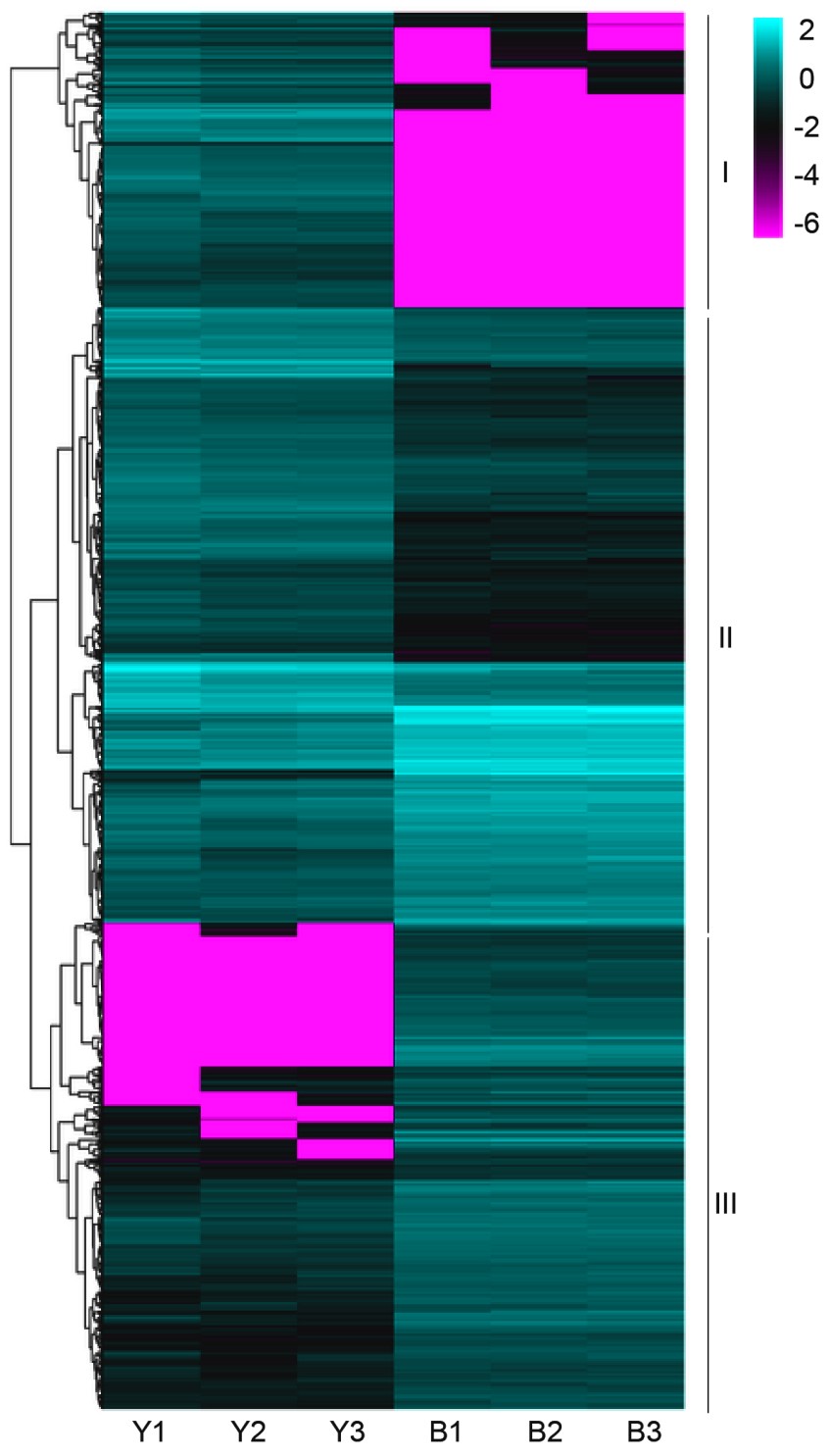

**Figure 4** **A heatmap of the expression profiles of differentially expressed genes (DEGs).** The expression level was normalized using the FPKM method, and the color scale bar represents this. Y1, Y2 and Y3 correspond to three biological replications of YG1. B1, B2 and B3 are the three biological replications of B100. DEGs with the same or similar expression pattern were clustered.

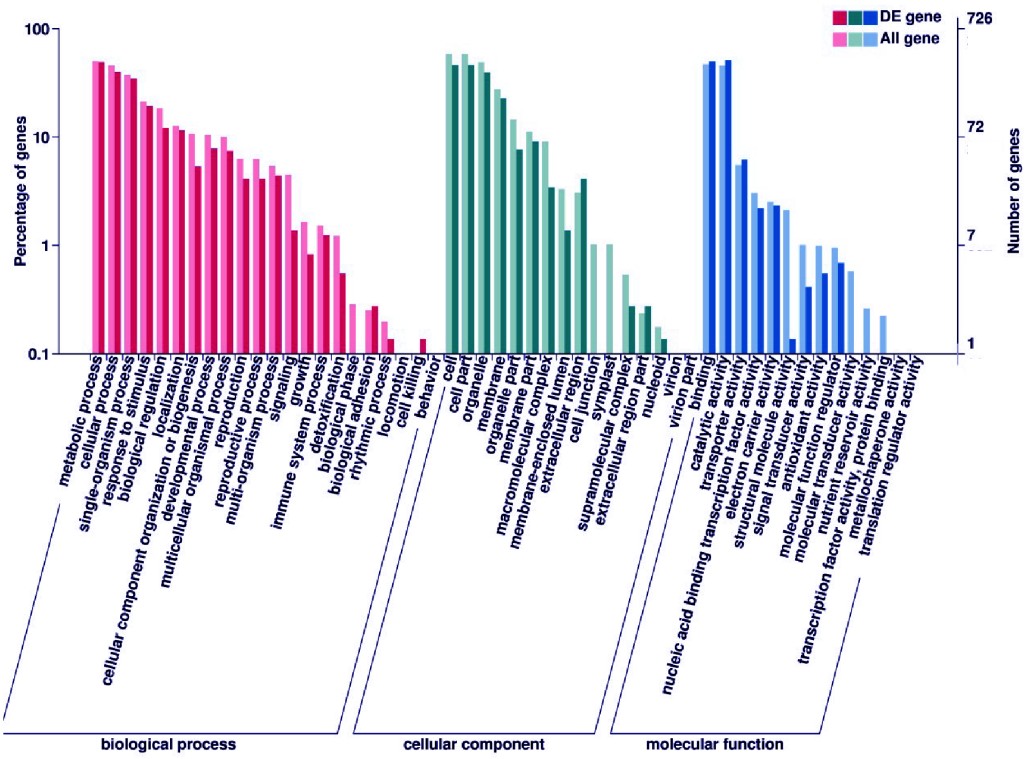

**Figure 5 GO enrichment analysis of DEGs.** The percentage of DEGs enriched GO terms of the biological process, cellular component, and molecular function are shown. The detailed gene information of GO enrichment analysis was displayed in Table S8.

scavenged in a timely manner and anthocyanins play an important role in this process (*Naing & Kim, 2021*).

Recent research has focused on constituents (*e.g.*, proteins and secondary metabolites) with beneficial effects on human health (*Sachdev, Goomer & Singh, 2021*). As natural pigments, anthocyanins are not only accumulated in flowers, fruits and seeds to give them brilliant colors, but they are also deposited in the vegetative organs such as leaves and stems. These pigments have protective and defensive roles. Our study showed that the anthocyanin accumulation in the purple-leaved variety of foxtail millet was significantly higher than that in green-leaved variety, and that the antioxidant capacity was also higher in the purple-leaved variety. These results suggested that the purple-leaved variety may improve stress resistance through the accumulation of more anthocyanins. Therefore, we can analyze the differences in resistance to adversity between purple- and green-leaved varieties to explore the relationship between anthocyanin accumulation and stress responses in foxtail millet. Through previous studies of germplasm resources, we identified approximately 100 purple-leaved varieties of foxtail millet. Some of these showed enhanced resistance to abiotic stress. There may be a potential relationship between the secondary metabolites and their responses to abiotic stress. Our work is expected to reveal the regulatory mechanism

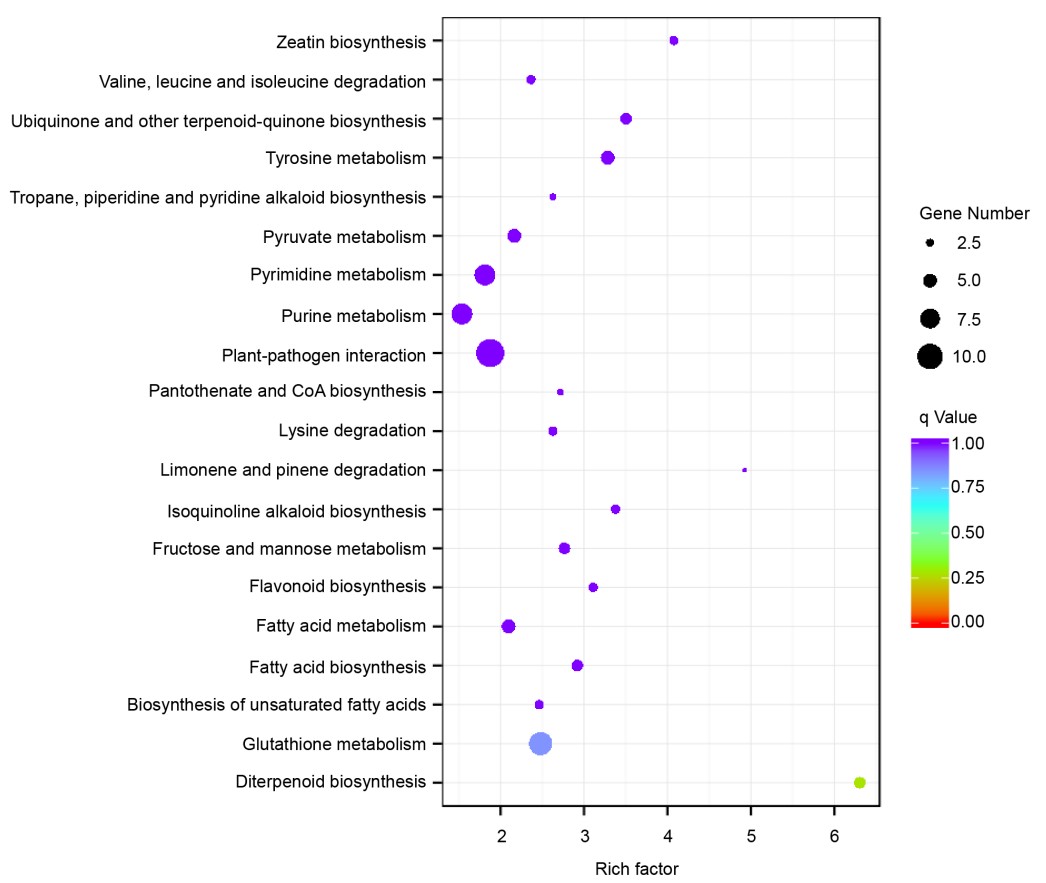

**Figure 6 Enriched KEGG pathways of the differentially expressed genes (DEGs).** KEGG pathways with corrected *P*-value < 0.05 were considered to be significantly enriched. The *y*-axis shows the description of the corresponding KEGG pathways. The top 20 KEGG pathways enriched in DEGs are shown.

of anthocyanin synthesis and the molecular basis of the high stress resistance of foxtail millet.

The distribution of anthocyanins in plants is not uniform, and variations are seen with plant species, growth stage and environments (*Guo et al., 2019*; *Zhang et al., 2019a*). Through observing leaves throughout the entire growth period, we found that leaves of the purple-leaved variety were only purple during the seedling and maturation stages but were green in other growth stages. This phenomenon may be due to the fact that plants need to enhance photosynthesis to provide the energy needed for growth at the jointing and heading stages.

Anthocyanin biosynthesis in plants is the results of a series of enzymatic reactions involving many enzymes. The first stage is phenylpropanoid biosynthesis from phenylalanine to p-coumaroyl-CoA. PAL and 4CL are the key enzymes of this stage. p-coumaroyl-CoA is then catalyzed to anthocyanins through flavonoid biosynthesis catalyzed by DFR and other key enzymes (*LaFountain & Yuan, 2021*). Genes encoding these enzymes are collectively referred to as structural genes (*Holton & Cornish, 1995*).

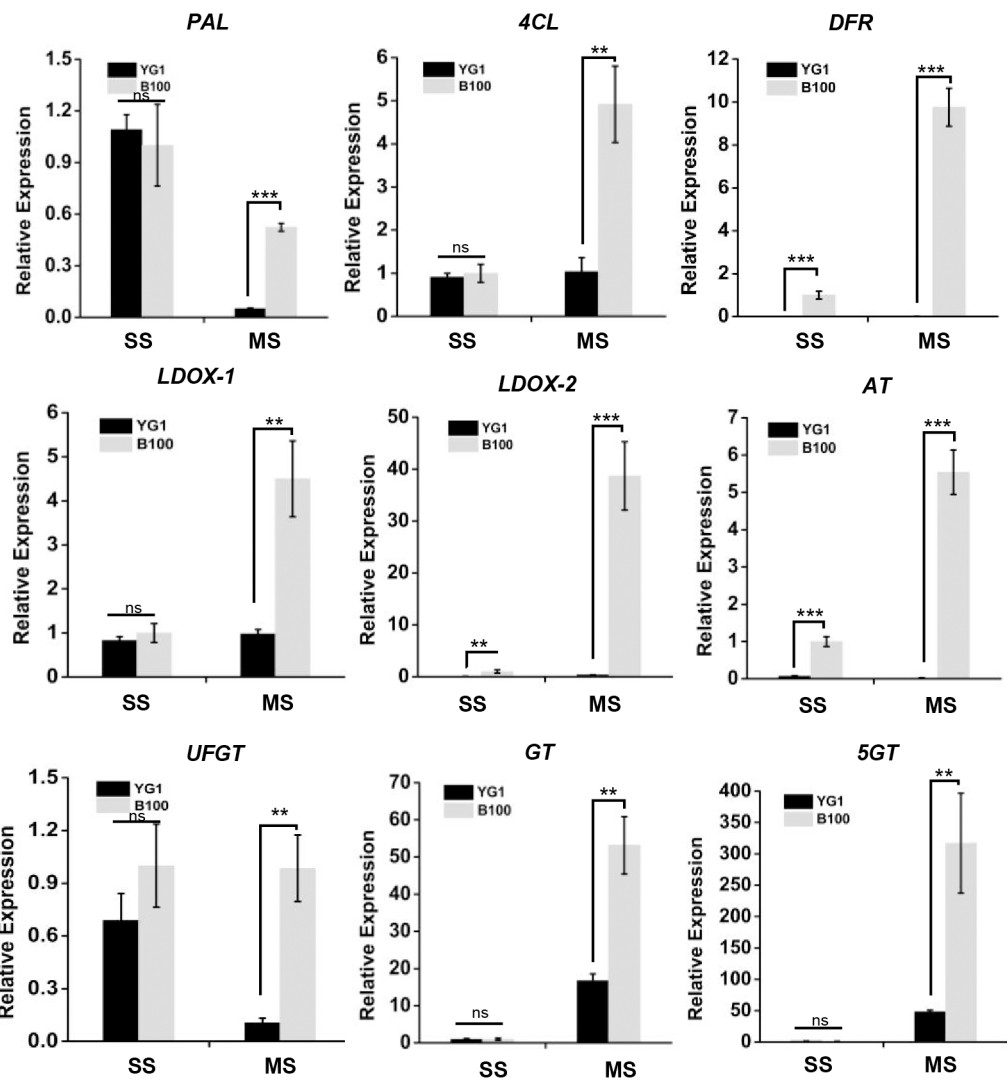

**Figure 7** **The transcriptional accumulation of enzyme encoding genes of anthocyanin biosynthesis.** The leaves of YG1 and B100 were sampled at 14 days old (seedling stage, SS) and 4 months old (maturing stage, MS). The relative expression levels of these genes were detected by quantitative real time PCR. *ACTIN* (5G464000, Seita) was used as an internal control. Two (**) and three (***) asterisks represent $P < 0.01$ and $P < 0.001$, respectively; ns represents no significant difference ( $n = 3$, ANOVA).

Through transcriptome analysis, we discovered that the expression levels of nine structural genes in the purple-leaved variety were higher than that in green-leaved variety. These results indicate that these genes were closely related to the accumulation of anthocyanin in foxtail millet. It has been demonstrated that anthocyanin biosynthesis was affected by structural genes and regulatory genes. The proteins encoded by regulatory genes affect the accumulation of anthocyanin by regulating the expression of the structural genes. The R2R3 MYB transcription factors, the basic-helix-loop-helix (bHLH) transcription factors, and the WD40-repeat proteins can control the expression of anthocyanin biosynthesis genes by forming a regulatory complex (*Xu & Zhang, 2015*). Researchers found that the purple

color of the pulvinus and leaf sheath of the foxtail millet paternal variety Shi-Li-Xiang (SLX) is caused by the interaction between the bHLH transcription factor PPLS1 and MYB transcription factor SiMYB85 promoting anthocyanin accumulation (*Bai et al., 2020*). Our transcriptome data also showed that the expression levels of some regulatory genes varied in purple- and green-leaved foxtail millet varieties. In future studies, we will explore the functions of these regulatory genes in anthocyanin accumulation in foxtail millet. B100 and Yugu1 have diverse genetic backgrounds and we will perform the whole-genome re-sequencing of B100 to discover more genetic variations associated with anthocyanin accumulation. There are some DEGs that have no relation with anthocyanin. However, some changes of gene expression between B100 and Yugu1 may be the result of anthocyanin up-regulation, such as DEGs that are mainly enriched in the plant-pathogen interaction. Anthocyanin accumulation has also been reported to be related to plant immunity (*Zhang et al., 2019b*).

China is one of the largest growers of foxtail millet, and its diversified natural and geographical conditions have produced rich germplasm resources. B100 is not a widely grown variety due to its low yield, however, it may have other functional uses. There are many arid and barren lands in developing areas that are not suitable for general crops. Varieties with high stress resistance like B100 are promising for their use in land utilization in severely arid regions. We have identified a great number of foxtail millet varieties which contains large amounts of anthocyanin and purple tissues (Table S9). The stems and leaves of millet with high anthocyanin content can be used as high-quality forage grass to enhance the economic value of animal products. However, we have not found any variety that accumulated abundant anthocyanin in the endosperm. We will further explore the differentially expressed genes, analyze the tissue specificity of their expression, and promote the breeding of varieties of foxtail millet with high anthocyanin grains to meet the health needs of humans.

## CONCLUSIONS

Foxtail millet is characterized by strong stress resistance and a rich nutritional content, however, the mechanisms of metabolites synthesis and stress resistance regulation are not clear. The identification and annotation of DEGs through RNA-sequencing suggested nine structural genes that played important roles in regulating the accumulation of anthocyanin. As the secondary metabolite, anthocyanin plays an important role in regulating a plant's response to stress. The research on the regulatory network of anthocyanin synthesis will provide a theoretical basis for high quality and highly stress-resistant crop breeding.

### Funding

This work was funded by the National Natural Science Foundation of China (General Program, 32070366 and 31971906), the Shanxi Province Science Found for Excellent Young Scholar (201901D211382), and the Scientific and Technological Innovation Programs of

Higher Education Institutions in Shanxi/ STIP (2021L118). The funders had no role in study design, data collection and analysis, decision to publish, or preparation of the manuscript.

## Grant Disclosures

The following grant information was disclosed by the authors:
National Natural Science Foundation of China: 32070366, 31971906.
Shanxi Province Science Found for Excellent Young Scholar: 201901D211382.
Scientific and Technological Innovation Programs of Higher Education Institutions in Shanxi/ STIP: 2021L118.

## Competing Interests

The authors declare there are no competing interests.

## Author Contributions

- Yaofei Zhao conceived and designed the experiments, performed the experiments, analyzed the data, prepared figures and/or tables, authored or reviewed drafts of the article, and approved the final draft.
- Yaqiong Li conceived and designed the experiments, performed the experiments, analyzed the data, prepared figures and/or tables, authored or reviewed drafts of the article, and approved the final draft.
- Xiaoxi Zhen performed the experiments, analyzed the data, prepared figures and/or tables, and approved the final draft.
- Jinli Zhang performed the experiments, analyzed the data, prepared figures and/or tables, and approved the final draft.
- Qianxiang Zhang performed the experiments, analyzed the data, prepared figures and/or tables, and approved the final draft.
- Zhaowen Liu performed the experiments, analyzed the data, prepared figures and/or tables, and approved the final draft.
- Shupei Hou performed the experiments, analyzed the data, prepared figures and/or tables, and approved the final draft.
- Yuanhuai Han conceived and designed the experiments, authored or reviewed drafts of the article, and approved the final draft.
- Bin Zhang conceived and designed the experiments, authored or reviewed drafts of the article, and approved the final draft.

## Data Availability

The data is available at NCBI SRA: PRJNA777600.

## Supplemental Information

Supplemental information for this article can be found online at http://dx.doi.org/10.7717/peerj.14099#supplemental-information.

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
