# Peer review of "Uncovering the mechanism of anthocyanin accumulation in a purple-leaved variety of foxtail millet (Setaria italica) by transcriptome analysis"

_PeerJ, doi:10.7717/peerj.14099_

## Round 0.1 · original submission · Minor Revisions

Please address the queries raised by the reviewers. Elaborate on the methods for the identification of DEG. Also, provide appropriate titles for all the figures and tables. The authors are also requested to include more information about the genes involved in anthocyanin production in the introduction and discussion sections of the manuscript.

·

Basic reporting

In this manuscript, Zhao et al. investigated the anthocyanin accumulation in a purple-leaved foxtail millet strain named B100. They started with the measurement of anthocyanin in leaves and then focused on the benefits of accumulating anthocyanin: rendering better stress resistance to the plants. Furthermore, the transcriptomic analysis highlighted the up-regulation of anthocyanin biosynthesis pathway genes, which can be validated by RT-qPCR. Overall, this study is well-structured with a professional presentation of the figures and results.

Experimental design

This study aims to provide a comprehensive analysis of B100, focusing on the mechanism and advantage of elevated anthocyanin synthesis. This research is within the aims and scope of PeerJ. In short, the authors found significant changes of anthocyanin in B100, compared to the Yugu1 strain. High-throughput sequencing and RT-qPCR validation revealed nine key genes involved in anthocyanin generation.
I have a few comments:
1, As mentioned in the RNAseq analysis, there are over 500 genes up- or down-regulated. However, only nine genes were finally identified for anthocyanin synthesis. What do the majority of significantly changed genes do? I noticed that the GO enrichment results indicated multiple functions. Are these changes the consequence of anthocyanin up-regulation?
2, The methods for RNAseq analysis were not clear enough. For example, “FDR<0.01 and Fold Change≥2” was used to identify significantly changed genes. How was the FDR calculated? What statistical test or software was used?
3, “Cluster analysis of these genes was conducted by Heatmap”. Heatmap was a way to visualize the clustering results, but clustering itself requires other methods, such as k-means, hierarchical clustering, etc. Please revise.

Validity of the findings

1, It is not clear to me whether anthocyanin accumulation elevated stress resistance, or the synthesis of anthocyanin is merely a consequence of sensitivity to stress. For example, under the same condition, Yugu1 won’t sense the stress, but B100 does.
2, Based on the transcriptomic analysis, thousands of genes changed between B100 and Yugu1. These changes cannot only be attributed to the expression changes, because the genetic differences need to be noticed. Have any genetic analyses (SNP, whole-genome sequencing, eQTL, etc.) been done so far (including results from other groups)? I suggest that the authors further elaborate on this in the discussion section.
3, Although B100 has better resistance, does it alter the taste or have any drawbacks? Will B100 produce higher economic value? A little more discussion is expected.

Overall, this manuscript presents solid data showing the mechanisms of anthocyanin biosynthesis, from phenotypic measurement to molecular level investigation. I suggest a minor revision.

Additional comments

In Figure 4, in my view, cluster III can be further divided into two clusters, with one having significantly changed genes, and another having moderate changes. What were the criteria to assign the three clusters?

Reviewer 2 ·

Basic reporting

Authors have identified a purple and green variety of foxtail millet and used them to study the morphological, metabolic and transcriptomic differences between the two varieties. They have identified 9 genes that were significantly upregulated in the purple variety B100 in comparison to the green variety YG1 that could possibly be involved in the higher anthocyanin production in this variety.

The authors have provided all the necessary raw data files, figures and tables for verification of their experiments. I request the authors to provide appropriate title for all the figures and tables along with footnotes describing any abbreviations used in them. The article also needs a through grammatical and spell-check.

I also recommend the authors to include more information about the genes involved in the anthocyanin production in the introduction and discussion sections of the manuscript.

Experimental design

The experimental design is appropriate for the current study. However, authors need to provide appropriate references for the methods described in the "methods" section. They should also include a section about the statistical method used for all their experiments.

Validity of the findings

Additional comments and suggested are mentioned in the edited version of this manuscript.

Annotated reviews are not available for download in order to protect the identity of reviewers who chose to remain anonymous.

Reviewer 3 ·

Basic reporting

The authors have addressed the importance of anthocyanin and its role in response to stress in Foxtail millet. The authors have referred relevant papers on the subject area and have laid a clear hypothesis to test using transcriptome data. The manuscript is written clearly, and flow of different sub-sections are appropriately defined.

Experimental design

1. The authors have caried out transcriptome an analysis comparing B100 and Yugu 1 varieties of the plant. It will be nice, if the authors could provide more details about the methods (alignment/mapping software’s) and parameters used in this study.
2. The authors have not detailed about the package or methods used to generate DEGs and how enrichment analysis was carried out. Like, which methods was used for enrichment, how the gene sets were identified, how the background genes were defined, how significant pathways were identified, p-values ect. Therefore, I urge the authors to provide more details and sequential steps followed in the methods section.
3. The authors have carried out clustering analysis using the DEGs, but the utility of these clusters and mechanistic background of genes are not well addressed. So, it will be great if the authors could provide more insights based on the current findings.
4. The authors have finally used 9 genes for further analysis, but it is not clear how these 9 genes were selected except that they were in the flavonoid biosynthesis. It will be nice to provide more details on the filtering step.

Validity of the findings

The findings from this study are interesting.

Additional comments

No comments

---

## Round 0.2 · Minor Revisions

The authors have addressed all the comments and suggestions made earlier and their responses are satisfactory. A few minor corrections are suggested by Reviewer 2, in the edited version of the manuscript.

In addition, Gerard Lazo, one of the Section Editors has made the following request for edits:

"There is a need to connect the sequence with the annotation terms; simply providing binned data without any context is basically meaningless. There are references to a reference genome, but there are no links to the data nor is the release version of the database mentioned. For instance, when mentioning a gene like Seita,1G. . . where would that sequence be obtained from and how can it be seen by the reader. A better job of providing navigation for the reader is needed.

There is a deficiency in labeling the annotation terms; for instance, the textual version of GO: terms is exacting and the GO:123456 indicators need be included somewhere, like the supplemental table alongside the gene ID and gene names and included in the Figure 5. As long as genes can be tracked in the supplemental table and identified to their DEG status then the histogram may suffice. Likewise there was no mention of the raw data being provided for inspection or through a third party resource like the NCBI SRA or the like. This is important since the study focused on so few secondary metabolism sequences and may have neglected others that may be followed-up upon.

There were a few places where a citation about a method would suffice. In the markup (forwarded separately) suggested edits were detected in lines 23, 122, 147, 149, 152, 163, 225, 286, 332. Forecasting future studies rarely comes to fruition; it is always best to provide something tangible. A germplasm list with a possible spread of phenotypes observed may provide some foundation that there are resources available, and that other researchers may collaborate if the reporting lab does not fulfill their promise; considering the breadth of samples screened it would be of extreme curiosity to know how the selected germplasm compared within the population.

In the current form the manuscript appears to have some deficiencies which need to be attended to. It does appear interesting, but requires a bit more navigation to be allowed to the prospective reader. I recommend further revision"

Please can you address all these issues.

·

Basic reporting

In this revised version, the authors have successfully addressed all the concerns I pointed out in the first round of review. I agree to accept this work for publication.

Experimental design

With the revision, all experimental designs meet the quality of PeerJ.

Validity of the findings

I don't have more concerns about the findings.

Reviewer 2 ·

Basic reporting

Authors have addressed all the comments and suggestions made earlier and their responses are satisfactory. There are few minor corrections suggested in the edited version of the manuscript.

Experimental design

No comments

Validity of the findings

No comments

Annotated reviews are not available for download in order to protect the identity of reviewers who chose to remain anonymous.

Reviewer 3 ·

Basic reporting

The authors have made great effort to address most of the concerning points, therefore would like to appreciate them.

Experimental design

The authors have addressed my comments efficiently.

Validity of the findings

No comments

Additional comments

No comments

---

## Round 0.3 · accepted · Accept

The authors have addressed the Reviewer's comments and queries. The manuscript can now be accepted.

Reviewer 2 ·

Basic reporting

Authors have addressed all the reviewers comments satisfactorily and the manuscript can be accepted for publication.

Experimental design

No comments

Validity of the findings

No comments

Additional comments

No comments